# MOC1 cleaves Holliday junctions through a cooperative nick and counter-nick mechanism mediated by metal ions

Danping Zhang[1], Shenjie Xu[2], Zhipu Luo ✉[2] & Zhonghui Lin ✉[1]

Holliday junction resolution is a crucial process in homologous recombination and DNA double-strand break repair. Complete Holliday junction resolution requires two stepwise incisions across the center of the junction, but the precise mechanism of metal ion-catalyzed Holliday junction cleavage remains elusive. Here, we perform a metal ion-triggered catalysis in crystals to investigate the mechanism of Holliday junction cleavage by MOC1. We capture the structures of MOC1 in complex with a nicked Holliday junction at various catalytic states, including the ground state, the one-metal ion binding state, and the two-metal ion binding state. Moreover, we also identify a third metal ion that may aid in the nucleophilic attack on the scissile phosphate. Further structural and biochemical analyses reveal a metal ion-mediated allosteric regulation between the two active sites, contributing to the enhancement of the second strand cleavage following the first strand cleavage, as well as the precise symmetric cleavage across the Holliday junction. Our work provides insights into the mechanism of metal ion-catalyzed Holliday junction resolution by MOC1, with implications for understanding how cells preserve genome integrity during the Holliday junction resolution phase.

Holliday junction (HJ) is a four-way structured DNA intermediate formed during the homologous recombination[1]. At the end of the process, HJs must be resolved into two DNA duplexes that can be readily repaired by the DNA ligase. Successful HJ resolution requires two stepwise incisions across the junction center, which is catalyzed by a group of structure-selective DNA endonucleases, namely HJ resolvases[2]. Based on the substrate specificity, HJ resolvases can be divided into two major categories[3]: (1) the canonical HJ resolvases, which includes RuvC[4–6], Hjc[4], Cce1 / Ydc2[5–8], GEN1[9–14], and MOC1[15,16]; and (2) the noncanonical HJ resolvases, such as SLX1-SLX4[17–20] and MUS81-EME1/2[21–23]. Canonical HJ resolvases introduce two symmetrical nicks across the junction center by functioning as homodimers, and exhibit strong structure- and sequence-specificities in HJ cleavage. For example, the *E. coli* RuvC prefers to cleave four-way junctions at the consensus 5'-$^A/_T$TT ↓ $^C/_G$-3'[24], while MOC1 specifically cleaves the

sequence of 5'-$^C/_T$C ↓ C-3' at the strand-exchanging point of HJ[15,16]. In contrast, the noncanonical HJ resolvases usually function as heterodimers, and make two asymmetric nicks nearby the junction center. In addition, the noncanonical HJ resolvases usually have a broad substrate specificity, such as three-way junctions, replication forks, and other branched structures including 5'-flaps and 3'-flaps[3].

Previous studies on RuvC[25,26] and GEN1[27] have revealed that the cleavage of HJ are operated through a nick and counter nick mechanism, which could result in failure of HJ cleavage when it occurs in the context of a branch-migrating junction, for instance, RuvC is believed to operate through forming a tripartite complex with the HJ branch migrator RuvAB[28,29]. It has been shown that RuvC accelerates the rate of second-strand cleavage by approximately 150-fold after the first cleavage, resulting a near simultaneous double incisions[25]. Such enhancement of the second-strand cleavage has been attributed to the

[1]College of Chemistry, Fuzhou University, Fuzhou 350108, China. [2]MOE Key Laboratory of Geriatric Diseases and Immunology, Institute of Molecular Enzymology, School of Biology and Basic Medical Sciences, Suzhou Medical College, Soochow University, Suzhou 215123, China. ✉e-mail: luozhipu@suda.edu.cn; zhonghui.lin@fzu.edu.cn

increase of junction flexibility following the first incision, yet the detailed molecular basis remains elusive.

The RNase H-like enzymes, including RuvC and MOC1, typically feature three or more conserved carboxylates, known as the DDE motif, in their active sites. The catalysis of these enzymes is generally dependent on divalent cations, with a preference for $Mg^{2+}$ and $Mn^{2+}$. For the metal ion-assisted catalysis, a general two-metal-ion mechanism was proposed by Steitz and Steitz[30], wherein one metal ion activates the hydroxyl nucleophile and the other stabilizes the pentacovalent intermediate. This proposal has gained extensive experimental supports over the past few decades. Remarkably, recent studies on the DNA polymerase η[31] and RNase H1[32] have shown that a third metal ion is required for DNA synthesis and RNA hydrolysis. This transient metal ion has been suggested to facilitate the products formation by coordinating either the leaving group during DNA synthesis[31] or the nucleophile in RNA hydrolysis[32]. Previous structural studies on the enzyme-substrate (ES) complex have suggested that HJ resolvases like RuvC[33] and MOC1[16] adopt a two-metal-ion catalysis mechanism. However, for crystallization purpose, these ES complex structures were mainly determined under non-reactive conditions either by the mutation of catalytic residues, or by the inclusion of $Ca^{2+}$ ions. Consequently, the precise mechanism of metal ion-catalyzed HJ DNA hydrolysis remains elusive.

In the present work, we conducted a crystallographic analysis to investigate the mechanism of metal ion-catalyzed HJ cleavage by MOC1. We captured the structures of the MOC1 in complex with a nicked HJ at various catalytic states, including the ground state, the one-metal ion binding state, the two-metal ion binding state. Moreover, we also identified a third metal ion that might aid in the nucleophilic attack on the scissile phosphate. These structural insights, combined with extensive biochemical analyses, shed light on the catalytic mechanisms through which the resolvases execute precise and efficient bilateral HJ cleavage.

## Results

### MOC1 cleaves HJ in a cooperative nick and counter-nick manner

Complete HJ cleavage requires two symmetric cleavages in the junction center through a nick and counter-nick mechanism. Previous studies on the HJ resolvases RuvC have shown that the rate of second strand cleavage is significantly accelerated by the first strand cleavage[25]. To investigate whether this effect also applies to MOC1, we designed a nicked HJ DNA substrate (nHJ), which contains a nick at one of the cleaving strands (Supplementary Table 1). In the gel-based HJ cleavage assay, MOC1 displayed significant enhancement in cleaving the nHJ compared to the intact HJ (iHJ) (Supplementary Fig. 1a). To quantitatively measure the rate of reaction, we employed a FRET-based HJ cleavage assay using the Cy3 and BHQ2 labeled HJ DNA as substrates (Fig. 1a, b). At concentrations of 88 nM, 175 nM and 350 nM, MOC1 cleaved iHJ with respective rates of $5.30 \pm 0.05 \times 10^{-5}$ nmol / min, $1.33 \pm 0.55 \times 10^{-4}$ nmol / min and $3.56 \pm 0.01 \times 10^{-4}$ nmol / min. For nHJ, the cleavage rates were $1.02 \pm 0.06 \times 10^{-4}$ nmol / min, $3.60 \pm 0.07 \times 10^{-4}$ nmol / min and $6.52 \pm 0.04 \times 10^{-4}$ nmol / min, respectively. These results thus suggest that MOC1 processes HJ through a cooperative nick and counter-nick manner.

### Crystal structure of MOC1 in complex with a cleavage intermediate

To explore the structural mechanism governing the cooperative nick and counter-nick HJ cleavage, we next determined the crystal structure of MOC1 in complex with nHJ, representing an intermediate state following the first strand cleavage. The structure was solved in the presence of $Ca^{2+}$ ions, which do not support catalysis (Supplementary Fig. 1b), at a resolution of 2.1 Å (Fig. 2a and Supplementary Data 1). The HJ DNA is well-defined in electron density, showing a break at the designated site in the junction center (Fig. 2a, b). The overall structures of MOC1 and HJ are similar to that of previously determined MOC1/iHJ complex (Supplementary Fig. 2a), with a root-mean-square deviation

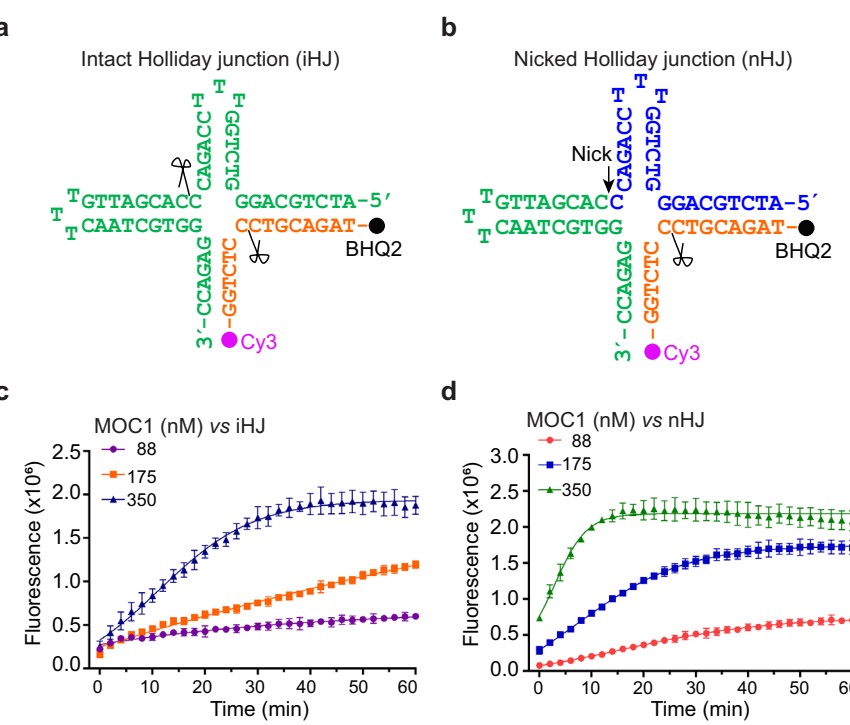

**Fig. 1 | FRET-based assessment of HJ cleavage by MOC1.** Schematic representation of the intact HJ (iHJ, **a**) and the nicked HJ (nHJ, **b**) DNA substrates used for the FRET-based HJ cleavage assay. The cleaving strands of HJs are labeled with Cy3 fluorophore and BHQ2 quencher as indicated. Arrow and scissor denote the nicked and the scissile sites, respectively. Real-time monitoring of the cleavage of iHJ (**c**) and nHJ (**d**) DNAs in the presence of 2 mM $Mn^{2+}$ and various concentrations of MOC1 at 30 °C. The fluorescent signal was recorded every 2 min. Values are mean ± SD of three replicates. Source data are provided as a Source Data file.

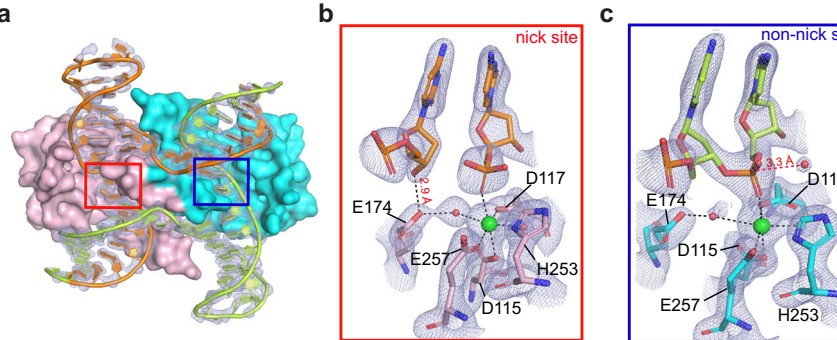

**Fig. 2 | Crystal structure of the MOC1/nHJ/Ca²⁺ complex. a** Cartoon diagram of the MOC1/nHJ/Ca²⁺ complex (PDB 8KFR). The two MOC1 subunits are colored in pink and cyan, respectively. HJ strands are colored in orange and green, and shown in ring representation with the superposition of the $2F_o$-$F_c$ electron density map contoured at 1.5 σ. **b, c** Zoomed-in view of the active sites of MOC1/nHJ/Ca²⁺ complex, superimposed with $2F_o$-$F_c$ maps. Ca²⁺ ions are shown in green spheres. Dashed lines indicate metal ion coordination or hydrogen bonds.

(RMSD) of 0.31 Å. This suggests that both MOC1 and HJ retain the structural integrity following the first strand cleavage.

A close examination in the catalytic center revealed that the active site on both the nick and the non-nick sites is occupied by a single Ca²⁺ ion (Fig. 1b, c). The coordination of Ca²⁺ involves the pro-Rp oxygen of the scissile phosphate, the carboxyl side chains of D115, D117 and E257, and two water molecules. In particular, the 3′-OH leaving group in the nick site accepts a hydrogen bond from E174. Notably, in the non-nick site, a water molecule was observed positioning 3.3 Å away from the phosphorus atom of the scissile phosphate, suggesting its potential role as a nucleophile. Overall, the coordination configuration of Ca²⁺ observed herein significantly deviates from the canonical two-metal-ion mechanism, explaining why Ca²⁺ ions do not support catalysis.

## Metal ion-triggered catalysis in crystals

The crystals of MOC1/nHJ complex offered an excellent foundation for analyzing HJ cleavage in crystals, as they ensured that cleavage would occur exclusively at one specific site, thus eliminating heterogeneity across the two active sites. Taking advantage of this, we next sought to examine Mn²⁺ ions-triggered HJ cleavage in crystals. For a detailed step-by-step process, please see the 'Method' section. First, to prevent spontaneous cleavage, the crystals of MOC1/nHJ complex were initially grown in the presence of Ca²⁺ ions, which were subsequently depleted by soaking in an EGTA containing solution. HJ cleavage was initiated by incubating the crystals in a solution supplemented with 10 mM Mn²⁺ ions for various periods. Following this, the crystals were briefly immersed in a cryoprotectant solution and rapidly cooled in liquid nitrogen for X-ray data collection. To examine metal-ion binding and accompanied structural changes during catalysis, we determined the crystal structures of MOC1/nHJ complex at a series of time points.

The structure of MOC1/nHJ complex in its ground state was determined at a resolution of 2.0 Å. The inspection of electron density map confirmed the depletion of Ca²⁺ ions, which were replaced by water molecules within both active sites (Fig. 3a). The overall configuration of catalytic site closely resembles that of our previous reported free MOC1[16], but differs significantly from the Ca²⁺ bound state, particularly in residues D115, D117, E174 and E257 (Fig. 3b). These observations thus suggest that the active site configuration has been reset to its initial ground state following the depletion of Ca²⁺ ions. We then proceeded with the reaction by soaking these crystals with Mn²⁺ ions. By analyzing the time-dependent changes of anomalous difference map, we observed stepwise Mn²⁺ binding within the catalytic site of MOC1 (Fig. 3c), which will be discussed further in the following sections.

The first metal ion rapidly appeared within both active sites for approximately 15 s after soaking in Mn²⁺ ions (Fig. 3c). It aligns with the A-site of the two-metal-ion mechanism proposed by Steitz and Steitz[30]. The single Mn²⁺ is coordinated by the pro-Rp oxygen of scissile phosphate, the carboxylate groups of D115 and E257, along with water molecules, forming an irregular coordination geometry (Fig. 3d). In comparison to the ground state, the binding of Mn²⁺-A triggers a notable conformational change of E257, while the other residues mostly remain unchanged (Fig. 3d).

The second Mn²⁺ emerged at the B-site by approximately 75 s (Fig. 3c). This sequential metal ion binding within the two sites might be attributed to the more solvent accessibility of the former compared to the latter (Supplementary Fig. 2b). Once the A site is occupied, the second metal ion has to compensate the charge-charge repulsion from metal ion A. This may explain why Mn²⁺-B has a relatively lower occupancy compared to Mn²⁺-A.

By around 180 s, Mn²⁺ ions had nearly fully occupied both the A and B sites (Fig. 3c). Both Mn²⁺ ions adopted a near octahedral geometry through coordination with the resolvase, HJ DNA substrate, as well as the water molecules (Fig. 3e). The distance between the Mn²⁺-A and Mn²⁺-B spans about 3.1-3.2 Å. In the nick site, the 5′-phosphate exhibits the characteristic inversion of its configuration and coordinates both metal ions, while the 3′-leaving oxygen is stabilized by Mn²⁺-B. Such coordination configuration is consistent with the post-cleavage state observed in the two-metal-ion mechanism[32]. In the non-nick site, Mn²⁺-A is coordinated by the pro-Rp oxygen of the scissile phosphate, residues D115, D117 and E257, and a water molecule, while Mn²⁺-B is coordinated by the O3′ atom of the scissile bond, residues D115 and E174, and two water molecules (Fig. 3e). Compared to the ground state, residues D117 and E257 undergo a side chain flip by approximately 90°, while D115 also rotates its side chain by approximately 45° (Fig. 3e). In contrast, H253, which does not involve in the coordination of both Mn²⁺ ions, almost remains unchanged during the reaction (Fig. 3e).

## Capturing the third metal ion

Interestingly, even though both Mn²⁺-A and Mn²⁺-B displays a near-perfect coordination geometry at approximately 180 s, only a small fraction of HJs were cleaved (Supplementary Fig. 2c). This intriguing observation led us to explore the possibility of a third metal ion that might be necessary for the catalysis to proceed. To capture this transient metal ion, we mutated a critical residue K229, in vicinity of scissile phosphate (Fig. 4a and Supplementary Fig. 3a, b). K229 is conserved across the RNase H superfamily that includes RuvC (Fig. 4b and Supplementary Fig. 3c). Its counterpart, K196 in RNase H1, has been demonstrated to play a crucial role in orienting the RNA substrate to capture transient cations for catalysis, and in aiding the removal of the cations to facilitate product turnover[32]. Consistent with its critical role

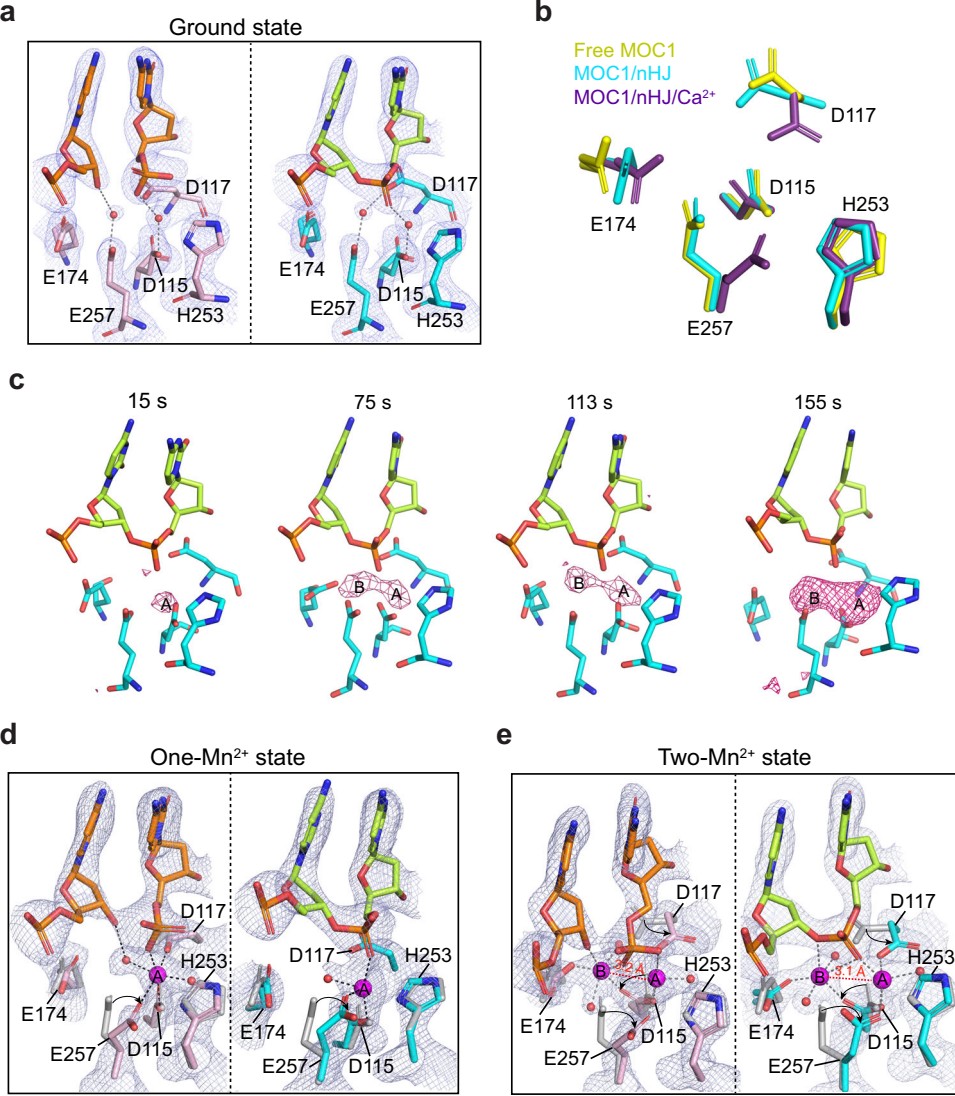

**Fig. 3 | Metal ion-triggered catalysis in crystals. a** Stick representation of the nick (left) and non-nick sites (right) of the MOC1/nHJ complex in the ground state (PDB 8KFS). Orange spheres indicate water molecules. Structures are superimposed with $2F_o$-$F_c$ electron density maps contoured at 1.5 σ. **b** Structural comparison of the active site of MOC1 in its free (yellow) (PDB 6IS9)[16], nHJ-bound (cyan) (PDB 8KFS), and nHJ and Ca²⁺-bound (magenta) (PDB 8KFR) states. **c** Stepwise metal ion binding revealed by the anomalous difference map contoured at 3.5 σ, superimposing onto the active site of the MOC1/nHJ complex in the ground state. Configuration of the nick (left) and non-nick sites (right) of the MOC1/nHJ complex in the one-Mn²⁺ (**d**) (PDB 8KFT) and the two-Mn²⁺ (**e**) (PDB 8KFU) bound states. Mn²⁺ ions are shown in magenta spheres. Structures are overlaid with $2F_o$-$F_c$ maps (1.5 σ). Conformational changes compared to the ground state (gray) are indicated by black arrows.

in HJ resolution, alanine substitution of K229 of MOC1 or the equivalent residue K119 of *Pseudomonas aeruginosa* (Pa.) RuvC significantly diminished HJ cleavage efficiency (Fig. 4c and Supplementary Fig. 3d).

We next crystallized MOC1^K229A in complex with nHJ. Crystals were soaked into Mn²⁺ containing solution over various durations prior to X-ray data collection. Following incubation the crystals with Mn²⁺ for about 180 s, we were able to detect the occupancy of metal ions at both A and B sites, as revealed by the anomalous difference map (Fig. 4d and Supplementary Fig. 3e). Notably, after approximately 600 s, a third Mn²⁺, referred to as Mn²⁺-C, began to emerge in the vicinity of H253, as confirmed by both the anomalous difference map (Fig. 4e) and the 2Fo-Fc electron density map (Supplementary Fig. 3f). Mn²⁺-C is coordinated by the Nε2 atom of H253 (2.2 Å), along with the solvent molecules. The point mutation of H253A significantly reduced HJ cleavage, this effect was not as drastic as seen with K229A and D115N mutations (Fig. 4f). We propose that Mn²⁺-C might synergize with Mn²⁺-A in activating the nucleophilic water, while K229 might neutralize the

negative charge of the pentavalent phosphorane intermediate, thereby facilitating product formation (Fig. 4g).

Interestingly, in certain species of RuvC, such as *P. aeruginosa* and *E. coli*, the equivalent residue of H253 is substituted with aspartic acid (Supplementary Fig. 4a). The MOC1 H253D mutant, at a concentration above 700 nM, retained significant activity in HJ cleavage, whereas the H253K mutant was nearly inactive even at a concentration of 1.4 μM (Supplementary Fig. 4b). This indicates that aspartic acid would play a similar role in the coordination of metal ion C.

## Metal ion-mediated allosteric regulation between the two active sites

One of the most intriguing findings from the above crystallographic analysis is the simultaneous appearance of metal ions at both the nick and the non-nick sites (Fig. 3d, e and Supplementary Fig. 5). This observation raises the possibility that the persistent metal ion coordination of the 5′-phosphate at the nick site after the first strand cleavage might be crucial for efficient cleavage of the second strand.

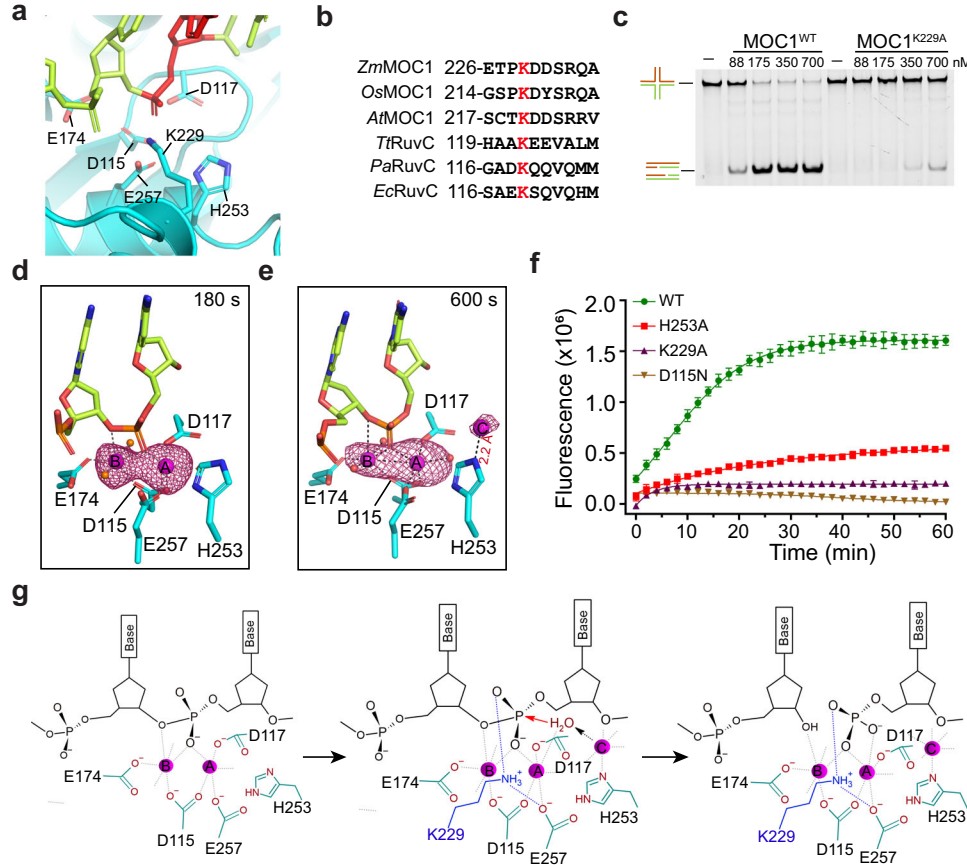

**Fig. 4 | Identification of a third metal ion for MOC1 catalysis. a** Cartoon representation of the non-nick site of MOC1/nHJ complex (PDB 8KFU). Key residues are shown in sticks and labeled. The scissile site on the HJ DNA is highlighted in red. **b** Sequence alignment across various species of MOC1 and RuvC proteins. *Zm, Zea mays; Os, Oryza sativa; At, Arabidopsis thaliana Pa; Tt, Thermus thermophilus; Pa, Pseudomonas aeruginosa; Ec, Escherichia coli.* **c** Effect of K229A mutation on the MOC1's activity of HJ cleavage by gel-based analysis. The reactions were carried out at 30 °C for 1 h, using 250 nM FAM-labeled HJ DNA, 2 mM $Mn^{2+}$ and indicated concentrations of MOC1. The experiment was independently repeated for three times with similar results. Metal ions binding at the non-nick site of the $MOC1^{K229A}$/nHJ complex following soaking in $Mn^{2+}$ containing solution for 180 s (**d**) (PDB 8KFV) and 600 s (**e**) (PDB 8KFW). The $Mn^{2+}$ ions depicted in magenta spheres are superimposed with anomalous difference map contoured at 3.5 σ. Water molecules are shown in orange spheres. Metal ion coordination and hydrogen bonds are indicated by dashed lines. **f** FRET-based real-time monitoring of nHJ cleavage by WT MOC1 or its variants. Reactions were carried out with 175 nM of MOC1 protein and 250 nM of nHJ in the presence of 2 mM $Mn^{2+}$. The fluorescent signal was recorded with 2-min intervals. Values are mean ± SD of three replicates. **g** Schematic diagram depicting the three-metal-ion catalysis mechanism for HJ cleavage by MOC1. Left: enzyme-substrate (ES) complex formation with metal ions A and B (magenta spheres); Middle: binding of metal ion C for substrate alignment and nucleophilic attacking, and the stabilization of pentavalent phosphorane intermediate by K229; Right: product formation and the binding of 5′-phosphate with K229. The red arrow indicates nucleophilic attack, while the dashed black arrow denotes the indirect activation of the nucleophilic water by metal ion C. Source data are provided as a Source Data file.

To test this hypothesis, we synthesized a nHJ lacking the 5′-phosphate at the nick site (nHJ-5′ PO4), which was anticipated to disrupt proper metal ion coordination. In agreement with the previous findings from RuvC[26], the absence of the 5′-phosphate at the nick site greatly compromised the cleavage of the second strand (Fig. 5a and Supplementary Fig. 6a). Moreover, when the nick was introduced at aberrant sites such as the $P_1$ and $P_{-1}$ positions, the cleavage efficiency of the nHJs were also markedly reduced (Fig. 5b and Supplementary Fig. 6b). In particular, the presence of a nick at the $P_{-1}$ site rendered the nHJ completely unreactive to cleavage (Fig. 5b and Supplementary Fig. 6b), likely due to the misalignment of the substrate within the active sites. These data suggest that precise alignment of both scissile phosphates with the metal ions within the active sites is critical for effective symmetrical cleavage. Owing to its intrinsic strand flexibility, the nHJ might achieve this alignment more rapidly than the iHJ, making it an optimal substrate for MOC1. To test this hypothesis, we compared the cleavage efficiency of iHJ and nHJ under various concentrations of $Mn^{2+}$ ions. We observed a dose-dependent responsive of $Mn^{2+}$ ions for the activation of HJ cleavage by MOC1, through both the FRET- and gel-based HJ

cleavage assays (Fig. 5c,d and Supplementary Fig. 6c). Consistent with our hypothesis, the half-maximal activation concentration for iHJ cleavage (0.10 ± 0.01 mM) is more than five times greater than that for nHJ (0.016 ± 0.0015 mM) (Fig. 5d).

## Discussion

Time-resolved *in crystallo* reaction is an important methodology to reveal detailed mechanisms of enzyme catalysis at atomic level. An important technique within this methodology is the use of diffusion-triggered reactions in crystals. This methodology is particularly useful to study the mechanism of metal ion-dependent enzymes that play fundamental roles in nucleic acid metabolism. The small size of $Mg^{2+}$ and $Mn^{2+}$ ions facilitates their rapid diffusion within the crystals. Using this approach, previous studies have successfully visualized reaction intermediates for DNA polymerase and ribonucleases, uncovering the significant roles of transiently bound metal ions in DNA synthesis and RNA hydrolysis[31,34]. In this study, we employed this methodology to study the mechanism of metal ion-catalyzed HJ resolution by MOC1. We successfully captured the sequential binding of metal ions and

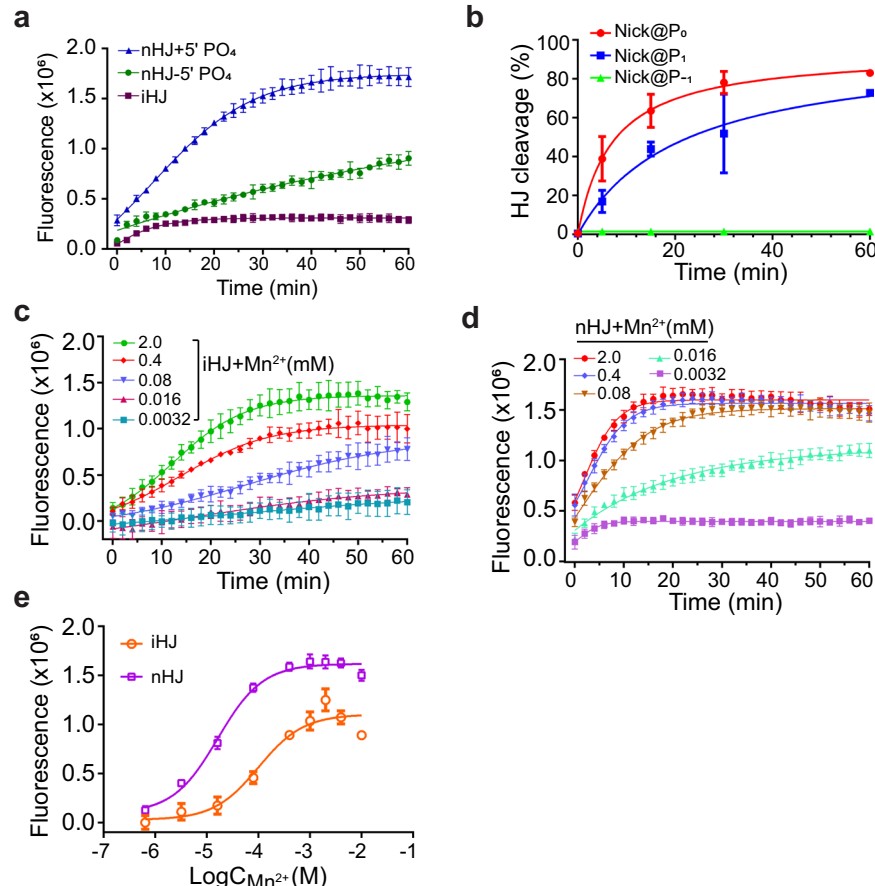

**Fig. 5 | The role of metal ions in the cooperative nick and counter-nick HJ cleavage by MOC1. a** Effect of the 5′ phosphate absence at the nick site on the cleavage of the second strand. The terms nHJ+5′ PO$_4$ and nHJ-5′ PO$_4$ refer to nHJ with and without the 5′ phosphate. Reactions were conducted under 175 nM MOC1, 2 mM Mn$^{2+}$ and 250 nM HJ substrates, using the FRET-based assay. Values are mean ± SD of three replicates. **b** Effects of aberrant cleavage at the first strand on the cleavage of the second strand. The terms nick@P0, nick@P1, nick@P-1 denote nHJ DNAs that harbors nicks at positions P0, P1 and P-1, respectively. Reactions were performed with 175 nM of MOC1, 250 nM of HJ DNA and 2 mM Mn$^{2+}$ at 30 °C

for 1 h. The data were the quantified results from the gel-based assay presented in Supplementary Fig. 6b. Values are means ± SD, $n = 2$ independent experiments. **c, d** Mn$^{2+}$-dose responsive activation of the HJ cleavage by MOC1. Reactions were carried out by mixing 175 nM MOC1 with 250 nM of either iHJ (**c**) or nHJ (**d**), in the presence of varing concentrations of Mn$^{2+}$. Values are mean ± SD of three replicates. **e** Sigmoidal dose-response curves of the stimulation HJ cleavage by Mn$^{2+}$. Data for iHJ and nHJ were obtained over reaction times of 30 min and 16 min, respectively. Values are mean ± SD of three replicates. Source data are provided as a Source Data file.

identified a third metal ion that potentially plays a crucial role in facilitating the nucleophilic attack on the scissile phosphate. Furthermore, our findings provided insights into how the HJ resolvases achieve precise and efficient bilateral HJ cleavages.

Consistent with the two-metal-ion mechanism proposed by Steitz and Steitz[30], our findings on the HJ cleavage by MOC1 support the crucial role of metal ion A in coordinating the nucleophilic water molecule, while metal ion B plays a vital role in stabilizing the 3′ leaving oxygen. The separation between Mn$^{2+}$-A and Mn$^{2+}$-B was found to be 3.1 to 3.2 Å, slightly less than the distances reported in the prior researches, such as 3.4 Å for the DNA polymerase η/DNA/dATP complex[34] and 3.5 Å for the RNase H/cleavage intermediate complex[35]. Notably, earlier studies have shown a considerable increase in the inter-metal distance following cleavage, with gaps expanding to 4.8 Å in the RNase H/product complex[35] and 5.5 Å in the GEN1/product complex[14]. These observations support the notion that movement of metal ions might be necessary in various stages of catalysis[35]. Although our research did not capture the post-cleavage state, we anticipate that a comparable increase in the separation between metal ions A and B.

Previous studies have highlighted the importance of a third metal ion for the efficient catalysis by DNA polymerase η[31] and RNase H[32], through its role in coordinating the leaving group and the nucleophile,

respectively. In this work, we noted that at 180 s, when both metal ions A and B achieved full occupancy, the majority of the HJ DNA remains uncleaved. This indicates that the presence of only metal ions A and B is insufficient for effective HJ cleavage. By mutating a critical residue K229, which appeared to stabilize the transient intermediate state, we were able to identify the third metal ion (metal ion C). This metal ion coordinates with the H253 as well as five water molecules. Although it does not directly coordinate the nucleophilic water, given its proximity to the phosphorus atom of the scissile phosphate (~6.4 Å), we propose that metal ion C might assist metal ion A to activate the nucleophilic water, likely through a water-mediated proton relay mechanism[36,37].

MOC1 shows remarkable structural and functional resemblance to bacterial RuvC proteins. Notably, in certain RuvC species, such as *P. aeruginosa* and *E. coli*, the equivalent residue of H253 is replaced by aspartic acid. This observation implies that aspartic acid could serve a role similar to that of histidine in the coordination of metal ion C. Consistent with this, our results showed that the H253D mutant of MOC1 demonstrated significant activity in cleaving HJ DNA. It worth noting that the catalytic site of RuvC lacks a direct analog to the residue D117 of MOC1. It would be interesting to study whether RuvC might employ a catalytic mechanism similar to that observed in MOC1.

Both MOC1 and RuvC, as canonical HJ resolvases, display pronounced sequence specificity in the cleavage of HJs. In the previous study, we have identified a base recognition motif (BRM) of MOC1 that mediates the sequence-specific cleavage of HJ[16]. For precise cleavage, the resolvases must scan the junction for a specific sequence, a process underscored by findings that HJs continue to undergo branch migration even when bound to the enzyme[38]. This requires the resolvases to make nearly simultaneous incisions to prevent asymmetric cleavage. How does MOC1 achieve this? Firstly, in line with the previous findings in RuvC[25], we observed a significant acceleration in the cleavage of the second strand following the initial strand cleavage. Such enhancement may be ascribed to the rapid metal ions alignment within the active sites, facilitated by the increased strand flexibility following the first strand cleavage. Secondly, we noted that the cleavage of the second strand was dramatically reduced when the first strand was cleaved at incorrect sites, thereby guaranteeing symmetric cleavage. Because the precise metal ion coordination with the 5′-phosphate at the nick site following the first strand's cleavage is crucial for the second strand's efficient cleavage, hence it is conceivable that improper cleavage at the first strand could lead to misalignment of the 5′-phosphate at the nick site, thereby hindering the cleavage of the second strand. Our work provides insights into the mechanism of metal ion-catalyzed HJ resolution by MOC1, with implications for understanding how cells preserve genome integrity during the HJ resolution phase, a crucial stage in the process of homologous recombination.

## Methods

### Protein expression and purification

For the protein expression and purification of MOC1 from *Zea mays* and its point mutants (aa. 109-271), the BL21 (DE3) pLysS bacterial cells expressing the Glutathione S-transferase (GST) tagged MOC1 recombinant proteins were disrupted by French Pressure (Union-Biotech, China) in lysis buffer containing 50 mM Tris-HCl, pH 8.0, 200 mM NaCl, 5% glycerol, 1 mM DTT and 0.1% Tween-20. The GST-tagged proteins were enriched on a GST Sepharose column (Senhui Microsphere Technology, China), and eluted with the elution buffer containing 20 mM Tris-HCl pH 8.0, 200 mM NaCl, 10 mM reduced glutathione and 1 mM DTT. Following the removal of GST tag using the home-made preScission protease, the untagged MOC1 proteins were further purified through the resource S cation-exchange column and Superdex 200 10/300 GL column (GE Healthcare).

### HJ DNA cleavage assay

For gel-based HJ cleavage assay, various amounts of MOC1 proteins and 250 nM of 6-carboxyfluorescein (FAM) labeled HJ substrate, prepared by combining four nucleotides (HJ-1 ~ 4, Supplementary Table 1) at 1:1:1:1 molar ratio, were mixed in the cleavage buffer containing 50 mM Tris 8.0, 50 mM NaCl, 10 mM $MgCl_2$ or indicated concentrations of $MnCl_2$, 5% (v/v) glycerol and 15% (v/v) DMSO. After incubation at 30 °C for 5 to 60 min, the reactions were stopped by incubating with 2 mg/ml proteinase K at 58 °C for 30 min. Subsequently, the reaction mixtures were separated on 10% polyacrylamide native gels, which were imaged and analyzed with the ChemDocTM Touch imaging system (Bio-Rad).

For FRET-based HJ cleavage assay, the iHJ DNA was prepared by annealing two oligonucleotides, iHJ-1 and iHJ-2, at 1:1 molar ratio, while nHJ DNA was prepared by mixing three oligonucleotides, nHJ-1, nHJ-2 and iHJ-2, at 1:1:1 molar ratio. The iHJ-2 was labeled with Cy3 and BHQ2 at the 5′ and 3′ ends, respectively. The sequences of these oligonucleotides were shown in Supplementary Table 1. For HJ cleavage, 250 nM of either iHJ or nHJ DNA was mixed with a specified concentration of MOC1 or its variants. The mixtures were incubated at 30 °C in a cleavage buffer containing 50 mM Tris-HCl (pH 8.0), 50 mM NaCl, 15% (v/v) DMSO, and 5% (v/v) glycerol. The reaction was initiated by the addition of varying concentrations of $Mn^{2+}$. Fluorescence intensities were recorded every 5 minutes for a total duration of 60 minutes, using a microplate reader (BioTek) with an excitation wavelength of 540 nm and an emission wavelength of 570 nm. Data analysis and graphing were performed using GraphPad Prism 6.0.

### Crystallization of the MOC1/nHJ complex

The nHJ DNAs for crystallization were prepared by annealing the following three oligonucleotides (oligo-a, b and c, Supplementary Table 1) at a molar ratio of 1:1:1. For the crystallization of MOC1/nHJ complex, the MOC1 protein (8 mg/ml) was pre-incubated with nHJ DNA substrate in the presence of 10 mM $CaCl_2$, at a molar ratio of 1:1.3. After a brief incubation on ice, the resulting protein-DNA mixture was used for crystal screening, using the commercially available screening kits from Hampton Research, Qiagen, and Molecular Dimensions. The crystals of MOC1/nHJ/$Ca^{2+}$ complex were obtained by mixing equal volumes of MOC1/nHJ complex and the precipitant containing 0.05 M $CaCl_2$, 0.1 M HEPES pH 7.0, 40% (v/v) polyethylene glycol (PEG) 200. The crystals of MOC1$^{K229A}$/nHJ complex were obtained with the precipitant containing 0.1 M MES pH 6.0, 35% (v/v) PEG 400.

### Metal ion-triggered catalysis in crystals

The $Mn^{2+}$-triggered catalysis of MOC1 in crystals was conducted as follows. Initially, to deplete $Ca^{2+}$ ions, the crystals of MOC1/nHJ/$Ca^{2+}$ complex, sized about 100 to 200 μm, underwent three washes with a pre-reaction buffer containing 25% (v/v) PEG 3350, 0.1 M HEPES pH 7.5, and 0.2 M KCl. This is followed by a 10-min incubation in a pre-reaction buffer supplemented with 0.5 mM EGTA. After another three washes with the pre-reaction buffer, crystals were transferred into a reaction buffer containing 10 mM $Mn^{2+}$, 25% (v/v) PEG 3350, 0.1 HEPES pH 7.5 and 0.2 M KCl, for various periods. Afterwards, the crystals were dipped briefly (5-10 s) in a cryoprotectant solution containing 20% (v/v) glycerol, 10 mM $Mn^{2+}$, 25% (v/v) PEG 3350, 0.1 HEPES pH 7.5 and 0.2 M KCl, and subsequently flash cooled in liquid nitrogen for X-ray data collection.

### X-ray diffraction data collection and structure determination

X-ray diffraction data were collected at 100 K on beamlines BL18U1, BL19U1 and BL02U1 at Shanghai Synchrotron Radiation Facility (SSRF), with the wavelength of 0.979 Å for native diffraction, and 1.50 Å for $Mn^{2+}$ anomalous diffraction, respectively, and processed with HKL2000[39] or XDS[40]. The initial phase of the MOC1/nHJ was obtained by molecular replacement using the structure of MOC1$^{D115N}$/iHJ/$Mg^{2+}$ complex (PDB ID: 6JRG)[16] as a searching model, with the Phaser-MR program in PHENIX[41]. Iterative model building and refinement were carried out with COOT[33] and PHENIX.refine. Data collection and refinement statistics were summarized in Supplementary Data 1. The anomalous differences were calculated by SHELXC[42]. The heavy atom sites and anomalous difference maps were generated by ANODE[43] and visualized in COOT. All structural figures in this study were generated with the program PyMOL (http://www.pymol.org/).

### Statistics & reproducibility

No statistical method was used to predetermine sample size. No data were excluded from the analyses; The experiments were not randomized; The Investigators were not blinded to allocation during experiments and outcome assessment.

### Reporting summary

Further information on research design is available in the Nature Portfolio Reporting Summary linked to this article.

## Data availability

The atomic coordinates and structure factors generated in this study have been deposited in the protein data bank (https://www.rcsb.org) under the following accession codes: 8KFR, 8KFS, 8KFT, 8KFU, 8KFV,

8KFW. Previously published coordinates and structures reused in this article can be found in the protein data bank under the following accession codes: 6JRG and 6IS9. Source data are provided with this paper.

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

## Acknowledgements

We thank the staff of BL18U1, BL19U1 and BL02U1 beamlines of National Facility for Protein Science in Shanghai (NFPS) at Shanghai Synchrotron Radiation Facility (SSRF), Shanghai, People's Republic of China, for assistance with X-ray data collections. This work is supported by the National Key R&D Program of China [2022YFA0806504] (Z. Luo) and the National Natural Science Foundation of China [31971222] (Z. Lin).

## Author contributions

Z. Lin conceived and designed the project. D. Zhang carried out all the cloning, biochemical and crystallization experiments. D. Zhang and S. Xu performed X-ray data collection. Z. Luo and Z. Lin determined the structures. Z. Lin wrote the manuscript with input of all authors. All authors read and approved the final manuscript.

## Competing interests

The authors declare no competing interests.
