## [Peer Review File · Nature Communications]

REVIEWER COMMENTS

Reviewer #1 (Remarks to the Author):

Danping Zhang, Shenjie Xu, Zhipu Luo & Zhonghui Lin. Time-resolved crystallography reveals metal ion-mediated cooperative nick and counter-nick mechanism of Holliday junction resolution by MOC1

This paper presents an interesting crystallographic study of catalytic metal ion binding into the active center of a Holliday junction-resolving enzyme from chloroplasts. This enzyme falls into the integrase superfamily of nucleases, that includes RuvC, Cce1 and RNaseH1. These enzymes in general exhibit significant sequence specificity for cleavage of four-way DNA junctions. I was curious about the statement that GEN1 is “non-canonical” and asymmetric. In fact GEN1 is very much a standard junction-resolving enzyme, very similar to those of lower organisms. I suggest the authors look at Freeman et al (*J. Molec. Biol.* 426, 3946-3959, 2014) that presents the biochemical analysis of GEN1 in detail. And perhaps the crystal structure of that enzyme too.

The structure of this enzyme has been determined by X-ray crystallography previously, and published in 2020. The new structures presented here study a nicked junction bound to the enzyme, so representing the complex at the half-way stage when the junction has been cleaved at one of the two sites. The resolution is good, and as far as I can see the statistics are good. I have no concerns about the structure determination. However, it would be good to see the experimental phasing map for the structures presented in the SI.

The main features of the new work is the analysis of metal ion binding. And particularly the time-resolved analysis after soaking in Mn²⁺ ions. I have a number of questions about this. How can we be sure (e.g. fig 2) we are really looking at electron density for metal ions? Mn²⁺ has strong anomalous diffraction at 0.91 Å that can be used to identify the Mn²⁺ ion positions. This is mentioned later in the paper, but how much is it used in general? Then I wonder about the timescale. The soaking is done for up to 10 mins, but how long does it take for the ions to diffuse through the lattice, and how homogeneous is the spot in the crystal from which the X-rays are scattering? Is a microfocus beam used? I notice that the gel shown in fig 2b shows product formation is almost complete by 600 s, yet there is a significant amount of product at zero time. The distance between the A and B ions is 3.1 Å. This is very close for two doubly-charged metal ions. The two ions in GEN1 are 5.5 Å apart (see Liu et al *Cell reports* 13, 2565–2575, 2015).

The authors make a big feature of the apparent third metal ion (C) appearing at late times of soaking. I am not convinced this is significant. First it is only observed when the critical lysine

(K229) is mutated to alanine, and second it only appears at 600 s. This is coordinated to the imidazole sidechain of H253; mutation of that histidine leads to impaired cleavage, but not a dead enzyme (fig 5C). But my understanding of this is hindered by trying to see what is being done. Successive lanes are not time, but rather labelled “mM”. This is not explained in the figure legend. In general the figure legends are too brief to convey a lot of necessary information on the experiments. On line 238 the authors state that the metal ion C would lower the pKa of the nucleophilic water molecule. I don’t understand this – it looks too far away to perturb the water molecule. A further point about the K229 (fig 4 a and b). It seems to me that the local resolution of the map doesn’t really allow clear conclusions about its position, and changes to that depending on the ionic environment.

Overall there are very few kinetic data in this paper, and no rates measured. The authors are studying an enzyme whose whole purpose is to accelerate a chemical reaction. They should report rates and errors.

Minor point. The term “in crystallo” is totally artificial and unnecessary. Please replace with the accepted term “crystallographic”.

Overall this study has some merit. I think it should be published, but needs some work to reach that point.

Reviewer #2 (Remarks to the Author):

This manuscript reports results indicating metal ion-catalyzed DNA hydrolysis in MOC1 and possible metal ion-mediated allosteric regulation between the two active sites. This work builds upon previous investigation of the group and likely to be of interest to the field. Overall the work appears to be sound and provides some interesting new mechanistic observations.

There are serious issues with this manuscript that should be addressed prior to publication. Notably the title of the paper is misleading.

"Time-resolved crystallography reveals metal ion-mediated cooperative nick and counter-nick mechanism of Holliday junction resolution by MOC1"

The crystallographic experiments performed are not time-resolved but are instead cryo-trapping experiments. This is an important distinction as time-resolved experiments suggest that there is a dynamic contribution to the data whereas cryo-trapping experiments are performed after freezing the sample in a particular state after initiating a reaction. The time-resolution of these experiments is very difficult to determine and it is more appropriate to refer to the incubation and freezing after various incubation times. It is unclear how long the crystals were incubated in the reaction buffer + cryoprotectant mixture prior to flash cooling.

The structures currently deposited in PDB are unreleased, please provide this data (pdb & mtz) to reviewers to verify the accuracy of model building and electron density maps.

It would be useful to provide electron density peak heights where metal ions are shown in structures.

Revisions

"The structure was solved in the presence of Ca²⁺ ions, which do not support catalysis ..." This statement should be justified with a short explanation.

Materials and methods:

Crystallisation - were the crystals grown in batch, sitting drop, hanging drop? What was the approximate size of the crystals used?

Time-resolved crystallography - The size of the crystals is important as it provides information about the diffusion time needed to soak in/out species from the crystals. How long were the crystals soaked during each step during the removal of metal ions, washing steps. Clearly state how long the crystals were incubated in reaction buffer and reaction buffer + 20% glycerol prior to flash cooling.

"The crystals collected at each time-point were flash frozen" This should read flash cooled.

X-ray diffraction data collection and structure determination - The temperature of data collection should be given.

How were the RMSD values between the various structures calculated?

Table 1

(I)/ σ (I) the sigma is missing. CC1/2 and CC* should also be included in table 1.

Figure 2c - Were Fo-Fo maps calculated between the metal free and each metal bound state? These would help to demonstrate the change in density in the metal binding site.

Figure 3 - Fo-Fc should be contoured no lower than 3.0 for a reliable indication of difference density.

Figure 4 - the density surrounding lysine 229 is weak especially surrounding the side chain nitrogen atom, providing accurate measurements of distances and drawing conclusions from these is not appropriate.

Figure 5 - why are the two difference maps contoured to different levels?

Reviewer #3 (Remarks to the Author):

What are the noteworthy results?

The authors describe a comprehensive (time-resolved) crystallography study on Holliday junction resolution by MOC1. They present a ground state metal free structure of Mco1, followed by soaking of Mn²⁺ illustrating the binding of one Mn²⁺ ion first as well as binding of the second Mn²⁺ after 180s soaking time. By mutating crucial K229 to alanine, followed by crystallisation of the mutant the authors demonstrated binding of a third Mn²⁺ with low occupancy by anomalous maps. By generating a few HJ analogues the authors provide initial data on metal ion-mediated allosteric regulation.

Will the work be of significance to the field and related fields? How does it compare to the established literature? If the work is not original, please provide relevant references.

The work shows in an elegant way new insights in the metal ion binding (in particular of a third Mn²⁺) on Holliday junction resolution by MOC1 and is of significant interest for the field.

Does the work support the conclusions and claims, or is additional evidence needed?

The work supports in general the conclusions of the author. Additional figures and more detailed discussions should improve the manuscript.

1) When exposing the crystals to EGTA how did you make sure that there was no EGTA left preventing your Mn²⁺ catalysed cleavage - in particular considering that the crystals seem to be fairly inactive? How reproducible is this? On how many crystals was this tested? Can you comment please. Thank you.

2) When subjecting the crystal to denaturing PAGE analysis how did you exclude that the cleavage is not just proceeding in solution - especially since you observe an on going cleavage on the PAGE analysis but not in crystallo? Is the PAGE data actually reflecting the in crystallo observations? If

yes, why is it not reflected in electron density maps? Have you analysed crystals on which diffraction data was collected by PAGE analysis after they are back from the synchrotron to have comparable data? Is it possible to repeat this for a few crystals?

3) The PDB validation reports suggest quite a bit of mismatches in the residue property plots. Are you confident that your models are all valid? Can you please show electron density maps for the substrate, in particular for all time points. So far only the anomalous maps of the metal are shown and it is crucial to also show the data for the substrate - especially in Figure 3 and 5. This can be done in additional panels or in additional Supplementary Figures in the SI. Thank you.

4) Is it possible to show a scheme revealing the isomorphous difference maps between the different time points to further validate the metal binding/change in occupancy in the time course? Thank you.

5) In Supplementary Figure 2 Can you please add information on which structures are compared including PDB codes. Thank you.

6) The figure legends are lacking in some cases crucial information. Can you please add comments on which occupancies the metals have been refined to in the deposited structures? Can you please also PDB codes to the figure legends to make it easier to follow. Thank you.

7) Can you please add the CC1/2 values to the data collection tables. Thank you.

Is the methodology sound? Does the work meet the expected standards in your field?

The work shows in an elegant way the binding of metals to Mco1 and supports an involvement of a potential third Mn²⁺ into catalysis. As described above the manuscript is missing crucial schemes/figures supporting the quality of the data in particular by missing figures on electron density maps of the substrate. This needs to be addressed with additional figures.

Is there enough detail provided in the methods for the work to be reproduced?

The experimental procedures are provided sufficiently. More thorough details are preferred for the figure legends. Thank you.

A point-by-point response to the referees' comments

Reviewer #1:

1. *This paper presents an interesting crystallographic study of catalytic metal ion binding into the active center of a Holliday junction-resolving enzyme from chloroplasts. This enzyme falls into the integrase superfamily of nucleases, that includes RuvC, Cce1 and RNaseH1. These enzymes in general exhibit significant sequence specificity for cleavage of four-way DNA junctions.*

Response: We are grateful to the reviewer for accurately summarizing the focus of our study.

2. *I was curious about the statement that GEN1 is “non-canonical” and asymmetric. In fact GEN1 is very much a standard junction-resolving enzyme, very similar to those of lower organisms. I suggest the authors look at Freeman et al (J. Mol. Biol. 426, 3946-3959, 2014) that presents the biochemical analysis of GEN1 in detail. And perhaps the crystal structure of that enzyme too.*

Response: We apologize for the oversight regarding the characterization of GEN1. We have updated our statement to classify GEN1 as a canonical Holliday Junction resolvase, and have cited the corresponding references (*J Mol Biol.* 2014, 426:3946-3959; *Cell Reports.* 2015, 13: 2565–2575) in the revised manuscript.

3. *The structure of this enzyme has been determined by X-ray crystallography previously, and published in 2020. The new structures presented here study a nicked junction bound to the enzyme, so representing the complex at the half-way stage when the junction has been cleaved at one of the two sites. The resolution is good, and as far as I can see the statistics are good. I have no concerns about the structure determination.*

Response: We appreciate the reviewer for the positive and encouraging comments on the structure determination of this study.

4. *However, it would be good to see the experimental phasing map for the structures presented in the SI. The main features of the new work is the analysis of metal ion binding. And particularly the time-resolved analysis after soaking in Mn²⁺ ions. I have a number of questions about this. How can we be sure (e.g. fig 2) we are really looking at electron density for metal ions? Mn²⁺ has strong anomalous diffraction at 0.91 Å that can be used to identify the Mn²⁺ ion positions. This is mentioned later in the paper, but how much is it used in general?*

Response: We understand the reviewer's concern about the electron density for metal ions. To address these concerns, we have replaced the *F_o-F_c* electron density map with the anomalous maps for each time point to specify the positions of Mn²⁺ ions (see Fig. 3c of the revised manuscript). These anomalous difference maps provide conclusive

evidence that the electron densities observed indeed correspond to Mn^{2+} ions, thereby reinforcing the validity of our analyses.

5. *Then I wonder about the timescale. The soaking is done for up to 10 mins, but how long does it take for the ions to diffuse through the lattice, and how homogeneous is the spot in the crystal from which the X-rays are scattering? Is a microfocus beam used? I notice that the gel shown in fig 2b shows product formation is almost complete by 600 s, yet there is a significant amount of product at zero time.*

Response: We appreciate the reviewer for the insightful comments on the crystal soaking with metal ions. First of all, it's important to clarify that, in Fig. 2b of the original version of the manuscript, the presence of the lower band at the zero-time point is attributable to the initial sample composition rather than being a product of enzymatic activity. The HJ DNA substrate used for crystallization contains a nick at one of the strand-exchanging points, representing the intermediate stage following cleavage at one of the two sites. Thus, the nHJ substrate consists of three strands with lengths of 8, 25, and 33 nucleotides, respectively. The 8 nt-strand was undetectable in the urea denaturing gel possibly owing to the limit of gel resolution.

The PAGE data shown in Fig. 2b do not correlate with the electron density maps in Fig. 2c of the original version of manuscript, because they were obtained under different experimental conditions. We found that the *in crystallo* HJ cleavage was optimal at pH 8.0. However, crystals under this condition decayed rapidly, resulting in poor X-ray diffraction. As a compromise, we opted to incubate the crystals at pH 7.5. This approach ensured better diffraction data quality but only achieved incomplete HJ cleavage in the crystals. To prevent any potential confusion, we have omitted the PAGE data from the revised manuscript.

Regarding to timescale, as far as we learned from the current study, metal ions can diffuse through the crystal lattice as quick as 15 seconds under current soaking condition (10 mM Mn^{2+} , 20°C). Similarly, in another pioneering work accomplished by Nakamura *et al.* on DNA polymerase η using the same methodology, they showed that the A site was completely occupied by Mg^{2+} within 40 seconds (*Nature*. 2012, 487: 196–201).

For X-ray data collection, each dataset was collected by using a single crystal ranging from 100 μm to 200 μm in size, and the beam size is typically set as 50 μm . The diffraction data processed by XDS revealed that the REFLECTING RANGE E.S.D., corresponding to the mosaicity, is 0.114~0.186, and the BEAM DIVERGENCE E.S.D., characterizing the Gaussian spot shape, is 0.022~0.025. This analyses suggest that the spot homogeneity is in a reasonable range.

6. *The distance between the A and B ions is 3.1 Å. This is very close for two doubly-charged metal ions. The two ions in GEN1 are 5.5 Å apart (see Liu et al Cell reports 13, 2565–2575, 2015).*

Response: We thank the reviewer for raising this important point. We noted that in the

structure of the GEN1/DNA complex determined by Liu *et al.* at a post-cleavage state, the separation between the two Mg^{2+} ions is 5.5 Å. Consistent with findings, an earlier study on the metal-ion catalysis mechanism in RNase H, where Nowotny *et al.* found that the metal ions are 4.8 Å apart in the product complex (*EMBO*, 25, 2006: 1924–1933). Intriguingly, this research also revealed narrower separations of 4.1 Å and 3.5 Å in the substrate and 'intermediate' complexes, respectively. This led to the hypothesis that a distance of less than 4 Å between metal ions A and B might be pivotal for pentacovalent intermediate formation. Supporting this notion, Nakamura *et al.* found a 3.4 Å gap between the Mg^{2+} ions A and B in the DNA polymerase η /DNA/dATP complex (*Nature*. 2012, 487: 196-201). Although our research did not capture the post-cleavage state, we anticipate a comparable increase in the separation of metal ions, supporting the notion that metal ions undergo movement during the catalytic process (*EMBO*, 25, 2006: 1924–1933). We have incorporated these discussions in the revised manuscript.

7. *The authors make a big feature of the apparent third metal ion (C) appearing at late times of soaking. I am not convinced this is significant. First it is only observed when the critical lysine (K229) is mutated to alanine, and second it only appears at 600 s. This is coordinated to the imidazole sidechain of H253; mutation of that histidine leads to impaired cleavage, but not a dead enzyme (fig 5C). But my understanding of this is hindered by trying to see what is being done. Successive lanes are not time, but rather labelled “mM”. This is not explained in the figure legend. In general the figure legends are too brief to convey a lot of necessary information on the experiments.*

Response: We appreciate the reviewer for this insightful point. At 180 s, when both metal ions A and B appeared, only a small population of HJ DNA was cleavage, suggesting that two metal ions are not sufficient to promote final product formation (Supplementary Fig. 2c). Inspired by the previous studies on RNase H (*NSMB*. 2018, 25: 715-721) and DNA polymerase η (*Science*. 2016, 352: 1334–1337), where the metal ion C has been shown to be required for product formation. Therefore, to capture the transient state at which metal ion C might exist, we mutated a critical residue K229 which seems to stabilize the transient state, and this led to the observation of metal ion C. It coordinates H253 as well as five water molecules. Such a weak binding affinity may explain why metal ion C is so transient to be detected.

The H253A mutation led to a significant reduction in HJ cleavage efficiency, while the D115N mutation entirely eliminated MOC1's catalytic function. These results were validated by our FRET-based HJ cleavage assay, which allowed us to monitor the processes in real time (Fig. 4f). Collectively, these data suggest that Mn^{2+} -C might synergize with Mn^{2+} -A in activating the nucleophilic water, thereby enhancing MOC1 catalytic efficiency.

We have incorporated these discussions and additional data in the revised version of manuscript, and have provided more detailed experimental information in the figure legends.

8. *On line 238 the authors state that the metal ion C would lower the pKa of the nucleophilic water molecule. I don't understand this – it looks too far away to perturb the water molecule.*

Response: We appreciate the reviewer for this insightful comments. Indeed, as the reviewer highlighted, metal ion C does not directly coordinate the nucleophilic water. Nonetheless, given its position within 6.4 Å from the phosphorus atom of the scissile phosphate, it might synergize with metal ion A to activate the nucleophilic water, likely through a water-mediated proton relay mechanism as proposed previously (*Phys Chem Chem Phys.* 2020, 22: 1534-1542; *J Biol Chem* 2002, 277: 5711-5714). We have included this discussion into the revised manuscript.

9. *A further point about the K229 (fig 4 a and b). It seems to me that the local resolution of the map doesn't really allow clear conclusions about its position, and changes to that depending on the ionic environment.*

Response: We agree with the reviewer that our previous statement on the position of K229 in Figs. 4a and 4b might be misleading, given that the ε-amino group of K229's side chain is largely disordered. K229 is conserved across the RNase H superfamily, including RuvC. Its counterpart, K196 in RNase H1, has been demonstrated to play a crucial role in orienting the RNA substrate to capture transient cations for catalysis, and in aiding the removal of the cations to facilitate product turnover. Our finding suggest that K229 of MOC1 might play a similar role in HJ hydrolysis. In addition, K229 may also serve to neutralize the negative charge of pentavalent phosphorane intermediate. We have revised the results and the corresponding statements in the revised manuscript.

10. *Overall there are very few kinetic data in this paper, and no rates measured. The authors are studying an enzyme whose whole purpose is to accelerate a chemical reaction. They should report rates and errors.*

Response: We appreciate the reviewer for this great suggestion. In response, we have employed a FRET-based assay using the Cy3/BHQ2-labeled HJ DNA as substrates (Fig. 1). This approach enabled us to monitor the enzyme's activity in real-time and determine both the rates and errors. We have re-evaluated the majority of our HJ cleavage experiments, and have incorporated these results into the updated version of our manuscript.

11. *Minor point. The term “in crystallo” is totally artificial and unnecessary. Please replace with the accepted term “crystallographic”.*

Response: We have replaced the term "in crystallo" with "crystallographic". Thank you for the suggestion.

12. *Overall this study has some merit. I think it should be published, but needs some work to reach that point.*

Response: We are grateful for the reviewer's recognition of the potential value in our study and the constructive comments. In response to these suggestions, we have conducted additional experiments and refined certain aspects of our manuscript. These revisions have significantly enhanced the quality of our paper.

Reviewer #2:

1. *This manuscript reports results indicating metal ion-catalyzed DNA hydrolysis in MOC1 and possible metal ion-mediated allosteric regulation between the two active sites. This work builds upon previous investigation of the group and likely to be of interest to the field. Overall the work appears to be sound and provides some interesting new mechanistic observations.*

Response: We appreciate the reviewer for the overall positive assessment of our work.

2. *There are serious issues with this manuscript that should be addressed prior to publication. Notably the title of the paper is misleading. "Time-resolved crystallography reveals metal ion-mediated cooperative nick and counter-nick mechanism of Holliday junction resolution by MOC1". The crystallographic experiments performed are not time-resolved but are instead cryo-trapping experiments. This is an important distinction as time-resolved experiments suggest that there is a dynamic contribution to the data where as cryo-trapping experiments are performed after freezing the sample in a particular state after initiating a reaction.*

Response: We have replaced the title with "Crystallographic Study Reveals Metal ion-mediated Cooperative Nick and Counter-nick Cleavage of Holliday Junction by MOC1". Thank you for the suggestion!

3. *The time-resolution of these experiments is very difficult to determine and it is more appropriate to refer to the incubation and freezing after various incubation times. It is unclear how long the crystals were incubated in the reaction buffer + cryoprotectant mixture prior to flash cooling.*

Response: We are grateful to the reviewer for highlighting this crucial aspect. In our study, the term "reaction time" specifically denotes the duration of incubation in the reaction buffer. For flash cooling, it typically takes 5 to 10 seconds of incubation in a cryoprotectant buffer. We have clarified this point in the 'Method' section of revised manuscript for better understanding.

4. *The structures currently deposited in PDB are unreleased, please provide this data (pdb & mtz) to reviewers to verify the accuracy of model building and electron density maps. It would be useful to provide electron density peak heights where metal ions are shown in structures.*

Response: In response to the reviewer's suggestion, we have provided the coordinate

files, including pdb and mtz, as supplementary information. Additionally, we have provided the *2Fo-Fc* electron density peak heights of corresponding metal ions in the Table.1 of the revised manuscript.

5. *"The structure was solved in the presence of Ca²⁺ ions, which do not support catalysis ..."* This statement should be justified with a short explanation.

Response: We thank the reviewer for raising this important issue. To address this point, we tested the HJ cleavage activity of MOC1 in the presence of various metal ions including Ca²⁺, Mg²⁺ and Mn²⁺, by using nicked HJ DNA as substrates. Our results indicated that Mg²⁺ and Mn²⁺, but not Ca²⁺, could stimulate the cleavage of nicked HJ by MOC1. This observation is consistent with our previous results on intact HJ (*Nat Chem Biol*, 15, 1241-1248, 2019). We have included these results in the revised manuscript (Supplementary Fig. 1b).

6. *Materials and methods:*

Crystallisation - were the crystals grown in batch, sitting drop, hanging drop? What was the approximate size of the crystals used?

Response: The crystals were grown using the method of hanging drop vapor diffusion. The size of the crystals utilized for in-crystallo reactions ranged approximately from 100 to 200 μm . We have included this description in the 'Methods' section of the revised manuscript.

7. *Time-resolved crystallography - The size of the crystals is important as it provides information about the diffusion time needed to soak in/out species from the crystals. How long were the crystals soaked during each step during the removal of metal ions, washing steps. Clearly state how long the crystals were incubated in reaction buffer and reaction buffer + 20% glycerol prior to flash cooling.*

Response: We apologize for the omission of specific details regarding the time-resolved crystallography procedures. First, to deplete Ca²⁺ ions, the crystals of MOC1/nHJ/Ca²⁺ complex, sized about 100 to 200 μm , underwent three washes with a pre-reaction buffer containing 25% (v/v) PEG 3350, 0.1 M HEPES pH 7.5, and 0.2 M KCl. This is followed by a 10-min incubation in a pre-reaction buffer supplemented with 0.5 mM EGTA. After another three washes with the pre-reaction buffer, crystals were transferred into a reaction buffer containing 10 mM Mn²⁺, 25% (v/v) PEG 3350, 0.1 HEPES pH 7.5 and 0.2 M KCl, for various periods. Afterwards, the crystals were dipped briefly (5-10 s) in a cryoprotectant solution containing 20% (v/v) glycerol, 10 mM Mn²⁺, 25% (v/v) PEG 3350, 0.1 HEPES pH 7.5 and 0.2 M KCl, and subsequently flash cooled in liquid nitrogen for X-ray data collection. We have updated the methods with detailed description in the revised manuscript.

8. *"The crystals collected at each time-point were flash frozen"* This should read *flash cooled*.

Response: The statement has been revised as suggested. Thank you!

9. *X-ray diffraction data collection and structure determination - The temperature of data collection should be given.*

Response: X-ray diffraction data were collected at 100 K. We have updated this information in the revised manuscript.

10. *How were the RMSD values between the various structures calculated?*

Response: The RMSD values between the various structures were calculated by aligning the structures to each other using PyMol software.

11. *Table 1: (I)/ σ (I) the sigma is missing. CC1/2 and CC* should also be included in table 1.*

Response: We apologize for the oversight. The missing sigma for (I)/ σ (I), as well as the values of CC1/2 and CC*, have now been incorporated into Table 1 of the revised manuscript.

12. *Figure 2c - Were Fo-Fc maps calculated between the metal free and each metal bound state? These would help to demonstrate the change in density in the metal binding site.*

Response: We agree with the reviewer that calculating the *Fo-Fc* maps between the metal free and each metal bound state would effectively capture the changes in density at the metal binding sites. However, our investigations revealed that the metal ion binding sites were frequently occupied by solvent molecules (Figs. 2b and 3a). This occupancy complicates the generation of difference maps to accurately reflect metal ion binding changes. To address this, and given the pronounced anomalous diffraction of Mn^{2+} ions at wavelengths between 0.978-1.50 Å, we have shown the anomalous difference maps in our revised manuscript (Fig. 3c). These maps specifically highlighted the exact locations and changes in occupancy of the Mn^{2+} ions, thereby providing more definitive insight into the binding dynamics. We hope this approach adequately addresses your concern and further strengthens our conclusion regarding the metal ion binding.

13. *Figure 3 - Fo-Fc should be contoured no lower than 3.0 for a reliable indication of difference density.*

Response: The maps depicted in Figure 3 are in fact the $2F_o-F_c$ electron density maps, for which contouring over 1.0 σ is considered to be reliable.

14. *Figure 4 - the density surrounding lysine 229 is weak especially surrounding the side chain nitrogen atom, providing accurate measurements of distances and*

drawing conclusions from these is not appropriate.

Response: We agree with the reviewer that the measurements of distances between K229 and the surrounding residues might be misleading, given that the ϵ -amino group of K229's side chain is largely disordered. In response, we have revised part of the statement in the revised manuscript.

K229 is conserved across the RNase H superfamily, including RuvC. Its analogue, K196 in RNase H1, is known to play a pivotal role in substrate orientation for catalysis by capturing transient cations and facilitating product turnover through cation removal. These insights lead us to propose that K229 in MOC1 may fulfill a similar function in the hydrolysis of HJ, potentially also contributing to the neutralization of the negative charge on the pentavalent phosphorane intermediate.

We have provided the related coordinate files, including PDB and MTZ files, available for the reviewers. We hope this additional information will address the reviewer's concerns on K229's role in HJ cleavage.

15. Figure 5 - why are the two difference maps contoured to different levels?

Response: We thank the reviewer for raising this question. The two difference maps are now contoured to the same level at 3.5σ .

Reviewer #3:

1. What are the noteworthy results? The authors describe a comprehensive (time-resolved) crystallography study on Holliday junction resolution by MOC1. They present a ground state metal free structure of Mco1, followed by soaking of Mn^{2+} illustrating the binding of one Mn^{2+} ion first as well as binding of the second Mn^{2+} after 180s soaking time. By mutating crucial K229 to alanine, followed by crystallisation of the mutant the authors demonstrated binding of a third Mn^{2+} with low occupancy by anomalous maps. By generating a few HJ analogues the authors provide initial data on metal ion-mediated allosteric regulation.

Response: We appreciate the reviewer for accurately summarizing the key findings of our study.

2. Will the work be of significance to the field and related fields? How does it compare to the established literature? If the work is not original, please provide relevant references. The work shows in an elegant way new insights in the metal ion binding (in particular of a third Mn^{2+}) on Holliday junction resolution by MOC1 and is of significant interest for the field.

Response: We appreciate the reviewer's overall positive assessment of the significance of our study.

3. Does the work support the conclusions and claims, or is additional evidence needed?

The work supports in general the conclusions of the author. Additional figures and more detailed discussions should improve the manuscript.

1) *When exposing the crystals to EGTA how did you make sure that there was no EGTA left preventing your Mn²⁺ catalysed cleavage - in particular considering that the crystals seem to be fairly inactive? How reproducible is this? On how many crystals was this tested? Can you comment please. Thank you.*

Response: We appreciate the reviewer's insightful inquiries regarding our methodology. In our study, we took two strategies to avoid the potential interference of EGTA on the subsequent Mn²⁺-catalyzed cleavage experiments. First, the EGTA concentration was set to 0.5 mM, which is significantly lower than the concentration of Mn²⁺ (10 mM) used in the cleavage experiments, thereby ensuring a surplus of Mn²⁺ ions available for catalysis. Second, to ensure the thorough removal of any residual EGTA prior to the cleavage reactions, the crystals were subjected to three washes with a pre-reaction buffer containing 25% (v/v) PEG 3350, 0.1 M HEPES pH 7.5, and 0.2 M KCl. Furthermore, to validate the reproducibility and reliability of our observations, we conducted experiments on at least 3 crystals for each catalytic state and consistently observed similar outcomes. Thank you for allowing us the opportunity to clarify these key aspects of our methodology. We have incorporated a detailed description of these methodologies in the revised manuscript's method section.

2) *When subjecting the crystal to denaturing PAGE analysis how did you exclude that the cleavage is not just proceeding in solution - especially since you observe an ongoing cleavage on the PAGE analysis but not in crystallo? Is the PAGE data actually reflecting the in crystallo observations? If yes, why is it not reflected in electron density maps? Have you analysed crystals on which diffraction data was collected by PAGE analysis after they are back from the synchrotron to have comparable data? Is it possible to repeat this for a few crystals?*

Response: We thank the reviewer for the concern regarding the possibility of cleavage occurring in solution during the denaturing PAGE analysis. For the gel-based examination of *in crystallo* HJ cleavage, we took a single crystal at each time point and dissolved it in a stop solution. This solution was composed of 50 mM EDTA and 2 mg/ml proteinase K, which was then followed by denaturing PAGE analysis. The inclusion of EDTA serves a critical role by chelating metal ions essential for the catalytic activity, while proteinase K eliminates the MOC1 protein. These steps ensure that the HJ cleavage observed in the PAGE analysis reflects the *in crystallo* reactions, rather than those potentially occurring in solution.

The PAGE data shown in Fig. 2b do not correlate with the electron density maps in Fig. 2c, because they were obtained under different experimental conditions. We found that optimal *in crystallo* HJ cleavage requires incubation at pH 8.0. However, crystals under this condition decayed rapidly, resulting in poor X-ray diffraction. As a compromise, we opted to incubate the crystals at pH 7.5, which ensured better diffraction data quality but resulted in incomplete HJ cleavage. To prevent any potential

confusion, we have omitted the PAGE data from the revised manuscript. We hope the reviewer could understand this situation, and that future advancement of X-ray crystallographic techniques would ultimately solve this issue.

3) The PDB validation reports suggest quite a bit of mismatches in the residue property plots. Are you confident that your models are all valid? Can you please show electron density maps for the substrate, in particular for all time points. So far only the anomalous maps of the metal are shown and it is crucial to also show the data for the substrate - especially in Figure 3 and 5. This can be done in additional panels or in additional Supplementary Figures in the SI. Thank you.

Response: We appreciate the reviewer for this constructive suggestion. In response, we have included the *2Fo-Fc* electron density maps for the substrate of each time point in our revised manuscript, specifically in Fig. 3 and Supplementary Fig. 3e,f. We have also provided the related coordinate files, including PDB and MTZ files, available for the reviewers. We hope this additional information will address the reviewer's concerns.

4) Is it possible to show a scheme revealing the isomorphous difference maps between the different time points to further validate the metal binding/change in occupancy in the time course? Thank you.

Response: We thank the reviewer for this suggestion. We agree that the isomorphous difference maps can provide valuable approach to validate changes in metal ion binding and occupancy over different time points. However, during our investigations, we found that the binding sites for metal ions were often occupied by solvent molecules (Figs. 2b and 3a). This occupancy complicates the generation of isomorphous difference maps to accurately represent metal ion binding changes. To address this, and in consideration of the anomalous diffraction properties of Mn^{2+} ions at wavelength of 0.978-1.50 Å, we have shown the anomalous difference maps in the revised manuscript (Fig. 3c). These maps specifically demonstrated both the precise locations and the variations in occupancy of the Mn^{2+} ions, offering a clearer insight into the binding events. We believe this adjustment adequately addresses your concern and enhances the manuscript's demonstration of metal ion dynamics.

5) In Supplementary Figure 2 Can you please add information on which structures are compared including PDB codes. Thank you.

Response: We have updated Supplementary Figure 2 to include information on the structures being compared, along with their corresponding PDB codes.

6) The figure legends are lacking in some cases crucial information. Can you please add comments on which occupancies the metals have been refined to in the deposited structures? Can you please also PDB codes to the figure legends to make it easier to follow. Thank you.

Response: We apologize for the oversight. We have updated the figure legends to

include crucial information about the refined occupancies of the metals in the deposited structures, as well as include PDB codes for easier reference.

7) *Can you please add the CC1/2 values to the data collection tables. Thank you.*

Response: We apologize for the oversight. The CC1/2 values have now been incorporated into Table 1 of the revised manuscript.

Is the methodology sound? Does the work meet the expected standards in your field?

Response: We thank the reviewer for raising this important question. Time-resolved *in crystallo* reaction is an important methodology to reveal detailed mechanisms of enzyme catalysis at atomic level. The realization that enzymes can remain active in crystalline forms goes back to 1926 with James B. Sumner's work on crystallized urease, which showed that crystallized urease can catalyze the breakdown of urea to ammonium and carbon dioxide (*The Journal of Biological Chemistry*. 1926, 69: 435–441).

An important technique within this methodology is the use of diffusion-triggered reactions in crystals. This strategy, made possible by the significant solvent space within protein crystals, allows for the diffusion of substrates, including large molecules like cytochrome c (*The Journal of Biological Chemistry*. 1983, 258: 5424–5427). Reactions can be paused for data analysis through flash-freezing or examined in situ using XFEL diffraction.

This methodology is particularly useful to study the mechanism of metal ion-dependent enzymes that play fundamental roles in nucleic acid metabolism. The small size of Mg²⁺ and Mn²⁺ ions facilitates their rapid diffusion within the crystals. Using this approach, previous studies have successfully visualized reaction intermediates for DNA polymerase and ribonucleases, uncovering the significant roles of transiently bound metal ions in DNA synthesis and RNA hydrolysis (*Nature*. 2012, 487: 196–201; *Science*. 2016: 352, 1334–1337).

By employing this methodology, in this study, we have demonstrated the sequential binding of metal ions in MOC1 during HJ cleavage, identifying a previously unrecognized third metal ion potentially assisting in the nucleophilic attack on the scissile phosphate. Moreover, we have also unveiled metal ion-mediated allosteric regulation between active sites, elucidating the enhanced efficiency observed in the second strand cleavage following the initial strand's cleavage.

We appreciate the chance to clarify this point, and have incorporated the discussion into the revised version of manuscript.

5. *The work shows in an elegant way the binding of metals to Mco1 and supports an involvement of a potential third Mn²⁺ into catalysis. As described above the manuscript is missing crucial schemes/figures supporting the quality of the data in particular by missing figures on electron density maps of the substrate. This needs*

to be addressed with additional figures.

Response: We appreciate the reviewer's acknowledgement of our work's significance and the insightful feedback provided. In response to the concerns raised, we have carried out further experiments and included additional figures, particularly electron density maps of HJ substrate. As a result, the quality of the manuscript has been significantly improved.

6. *Is there enough detail provided in the methods for the work to be reproduced? The experimental procedures are provided sufficiently. More thorough details are preferred for the figure legends. Thank you.*

Response: We thank the reviewer for this suggestion. We have provided more detailed descriptions in the figure legends, which should facilitate more comprehensive understanding of the results.

REVIEWERS' COMMENTS

Reviewer #1 (Remarks to the Author):

This is a paper that is basically sound, and will interest people working on the structural and mechanistic aspects of homologous genetic recombination. At this stage I recommend acceptance.

Reviewer #2 (Remarks to the Author):

The manuscript has been much improved by the inclusion of anomalous difference maps to justify the conclusions drawn about the appearance of a secondary and tertiary Mn⁺ ion with increased soaking times.

The authors have also added additional detail to the materials and methods section to clarify the timescales of the various soaking procedures and the size of crystals used in the experiment.

Supplementary figures have been included to justify the point raised about calcium ions not supporting catalysis.

There are some issues with modelling alternative conformations in some of the structures (e.g. ARG161A in 8KFV) and water molecules in a number of the structures, while these do not appear to affect the conclusions drawn and should not prohibit publication, I urge the authors to take greater care in future.

Reviewer #3 (Remarks to the Author):

The changes in the revised manuscript have significantly improved the manuscript. Thank you for providing the coordinates and electron density maps - the maps look convincing to me. In particular the addition of figures on the time dependent changes in the anomalous difference map reveal the

consecutive binding of the Mn^{2+} . All concerns and suggested changes have been addressed and I would recommend the manuscript for publication.

A point-by-point response to the referees' comments

Reviewer #1:

1. *This is a paper that is basically sound, and will interest people working on the structural and mechanistic aspects of homologous genetic recombination. At this stage I recommend acceptance.*

Response: We are grateful to the reviewer for the positive comments on our work.

Reviewer #2:

1. *The manuscript has been much improved by the inclusion of anomalous difference maps to justify the conclusions drawn about the appearance of a secondary and tertiary Mn²⁺ ion with increased soaking times. The authors have also added additional detail to the materials and methods section to clarify the timescales of the various soaking procedures and the size of crystals used in the experiment. Supplementary figures have been included to justify the point raised about calcium ions not supporting catalysis.*

Response: We are grateful to the reviewer for the positive comments on our work.

2. *There are some issues with modelling alternative conformations in some of the structures (e.g. ARG161A in 8KFV) and water molecules in a number of the structures, while these do not appear to affect the conclusions drawn and should not prohibit publication, I urge the authors to take greater care in future.*

Response: We thank the reviewer for the suggestion. We have corrected these issues and will take greater care in future.

Reviewer #3:

1. *The changes in the revised manuscript have significantly improved the manuscript. Thank you for providing the coordinates and electron density maps - the maps look convincing to me. In particular the addition of figures on the time dependent changes in the anomalous difference map reveal the consecutive binding of the Mn²⁺. All concerns and suggested changes have been addressed and I would recommend the manuscript for publication.*

Response: We are grateful to the reviewer for the positive comments on our work.